# Functional Evaluation of Splicing for Variants of Uncertain Significance in Patients with Inherited Retinal Diseases

**DOI:** 10.3390/genes12070993

**Published:** 2021-06-29

**Authors:** Margarita Mauro-Herrera, John Chiang, Bojana Radojevic, Lea D Bennett

**Affiliations:** 1Department of Ophthalmology, University of Oklahoma Health Sciences Center, Oklahoma City, OK 73114, USA; Margarita-MauroHerrera@ouhsc.edu (M.M.-H.); Bojana-Radojevic@ouhsc.edu (B.R.); 2Molecular Vision Laboratories, Hillsboro, OR 97006, USA; jchiang@mvisionlab.com

**Keywords:** inherited retinal disease, VUS, functional analysis, minigene assay

## Abstract

Inherited retinal diseases (IRD) comprise a heterogeneous set of clinical and genetic disorders that lead to blindness. Given the emerging opportunities in precision medicine and gene therapy, it has become increasingly important to determine whether DNA variants with uncertain significance (VUS) are responsible for patients’ IRD. This research was performed to assess the functional consequence of six VUS identified in patients with IRD. Clinical assessments included an ophthalmic examination, best-corrected visual acuity, and kinetic perimetry. Imaging was acquired with the Optos ultra-widefield camera and spectral domain optical coherence tomography (SD-OCT). Genetic testing was performed by Molecular Vision Laboratories. VUS that were predicted to alter splicing were analyzed with a minigene assay, which revealed that VUS in the genes *OPA1*, *CNGB1*, and *CLUAP1* altered spicing mechanisms. Due to emerging gene and cell therapies, these results expand the genotype-phenotype correlations for patients diagnosed with an IRD.

## 1. Introduction

Inherited retinal diseases (IRDs) have a prevalence of 1 per 3000 individuals and are among the leading causes of vision impairments that develop due to a genetic mutation [1]. IRD comprises a largely heterogeneous set of clinical and genetic disorders that lead to blindness. To date, 271 genes have been identified that cause IRD (www.sph.uth.edu/retnet/sum-dis.htm; accessed on 10 April 2021). Given the emerging opportunities in precision medicine and gene therapy, it is increasingly important to provide patients with a genetic diagnosis. Next generation sequencing (NGS) has advanced the detection of pathogenic mutations in 60–80% of patients with IRD [2]. Despite the need for a definitive genetic diagnosis, NGS frequently identifies DNA changes that are classified as variants of uncertain significance (VUS), leaving the genetic, clinical, or both diagnoses unresolved. Validating the functional consequence of VUS aids in determining whether it is the causative mutation for a patients’ IRD. Functional analysis of VUS impacts the standard for patient care, which will lead to earlier diagnosis, new treatments, and prevention of blindness.

Recently, RNAseq studies of the retina have shown that a large fraction of the human genome is alternatively spliced with about 50% of genes displaying various exon composition compared to the reference sequence [3,4,5]. Photoreceptors in particular express different exons that are not found in extra-retinal tissues, [6] signifying that alternative splicing can be cell-type specific. Although underestimated, it has been projected that 25% of all variants within exons influence splicing mechanisms [7]. Additionally, intronic variants can also alter splicing motifs, thereby disrupting splicing regulation. There are numerous variants that have not been characterized at the transcript level, especially those found in patients with IRD. Direct analysis of native transcripts is ideal, but it is not possible for genes specifically expressed in the retina. In such instances, a minigene assay (also called exon trapping) has been used for validating the predicted splicing defects associated with VUS [8,9]. By transfecting mammalian cells such as COS7 cells, splicing mechanisms can be evaluated by analyzing the transcripts expressed and processed by the cell’s inherent splicing machinery. In this study, we use a minigene in vitro assay to functionally test six VUS in the genes *OPA1*, *CNGB1*, and *CLUAP1* that were predicted to alter spicing mechanisms.

## 2. Materials and Methods

Informed consent was obtained prior to blood collection or clinical exams, which was approved by an institutional ethics review board and which adhered to the Declaration of Helsinki. Clinical exams were performed as previously described [10]. Briefly, the Optos^®^ camera (Optos PLC, Dunfermline, UK) was used to obtain pseudocolored and fundus autofluorescence (FAF) retinal images. The spectral domain optical coherence tomography (SD-OCT) images were obtained with the Spectralis Heidelberg retina angiography + OCT (Heidelberg Engineering, Inc., Franklin, MA, USA). Kinetic perimetry was performed using an Octopus900 (Haag-Streit AG, Köniz, Switzerland). For the Octopus900 kinetic exam, the field was mapped using spot sizes V-4e, III-4e, and I-4e at a speed of 4°/s. The Electro-Oculogram (EOG) was acquired using the UTAS SunBurst (LKC Technologies; Gaithersburg, MD, USA) according to ISCEV recommendations [11] and compared to the normal Arden ratios provided with the software.

Genetic testing was performed by a CLIA-certified genetic laboratory (Molecular Vision Laboratory, Hillsboro, OR, USA). Sequencing results were aligned to the reference transcript (Appendix A), with the first nucleotide position corresponding to the A of the ATG translation initiation codon. Variant nomenclature was reported in accordance with the recommendations of the Human Genome Variation Society [12] and interpreted according to the American College of Medical Genetics and Genomics (ACMG) and the Association for Molecular Pathology (AMP) [13,14]. In silico prediction of the functional consequence of the VUS was evaluated with four different algorithms embedded in the Alamut Visual version 2.15 (SOPHiA GENETICS, Lausanne, Switzerland) using default settings. The VUS were chosen for analysis because they were detected in our patient population and there was no functional evidence in ClinVar for these VUS. In vitro splice assays were performed following the previously described protocol [2,15]. Briefly, a genomic segment encompassing the variant of interest, along with flanking sequences, was PCR-amplified from patient genomic DNA (gDNA) and was cloned into the exon-trapping vector pSPL3. The primers (Appendix A) were designed [16] to encompass the region of interest along with ~150 base pairs (bp) upstream and downstream of the exon containing the VUS or away from the intronic variant and the nearest exon (Appendix A). The pSPL3 vector encoded two vector exons (V1 and V2) with functional splice sites. The amplicon was cloned between the vector exons at a multiple cloning site. Subjects were heterozygous for the splice VUS of interest and used for co-amplification of the wild-type (WT) and mutant (MUT) alleles. The resultant constructs in their WT and MUT versions were used to transfect eukaryotic cells (COS-7). Transfections were done using the Lipofectamine 3000 transfection reagent (Invitrogen, Waltham, MA, USA). A 1:2 ratio of plasmid DNA to Lipofectamine was used. COS-7 cells were incubated with a DNA plasmid-Lipofectamine complex for 6 h in an Opti-MEM^®^ Reduced-Serum Medium (Gibco-ThermoFisher, Waltham, MA, USA). After 6 h, the media was changed to DMEM (Dulbecco’s Modified Eagle’s Medium), supplemented with 10% FBS and a 1% antibiotic. Cells were collected 48 h after transfection. After transfection, the RNA was isolated, and cDNA was created. RT-PCR was performed with pSPL3 vector-specific primers [15] to ensure selective amplification of vector-derived transcripts. The minigene-derived transcripts were visualized by agarose gel electrophoresis. Subsequently, transcripts were isolated from the agarose gel and Sanger sequenced to delineate the exact splicing defect.

To quantify transcripts, a modified vector was created by subcloning an mCherry/Blasticidin sequence driven by the CMV (Cytomegalovirus) promoter between the pBR322_origin and the SV40_promoter of the pSPL3 vector. This vector was called mCherry_pSPL3. Transfected cells were positively selected by incubating cells with Blasticidin (5 µg/mL). Transfected cells were collected after 48 h in regular DMEM supplemented with 10% FBS and 1% antibiotic and plated at ~20% confluency in DMEM supplemented with 10% FBS and 7 ug/mL blasticidin. COS-7 cells treated with Lipofectamine 3000 but not plasmid were used as a control to follow cell viability with blasticidin. An equal number (1 × 10^5^) of cells per sample was used to make cDNA and perform RT-PCR. PCR products were visualized by agarose gel electrophoresis and images were analyzed with ImageJ [17].

Three different transformation clones (technical replicates) with WT or MUT plasmids were used for transfections, cDNA synthesis, and RT-PCR analysis. When possible, constructs from different individuals with the same VUS were used (biological replicates). Assays were performed two to three times with similar results for verification. Densitometry was used to measure band intensity and quantify transcripts made by the minigene assays. The percentage of transcripts was determined after subtracting the background on the images [18]. The value for the transcript of interest was divided by the total amount of transcripts (sum of all expressed transcripts) and multiplied by 100 to get the percentage. Measurements were repeated three different times by two individuals (MMH and LDB). Values were averaged (±SEM) and compared between WT and MUT using the paired two-tailed Student’s *t*-test. Significance was given as a *p*-value of less than 0.05.

## 3. Results

Patient #1 was diagnosed with an IRD at age 46. An ophthalmic exam at age 56 revealed symmetrical pathology between the eyes by imaging, psychophysical testing, and ophthalmic examination. The representative fundus photo of the right eye showed pigmentary and degenerative changes in the macula (Figure 1A). Geographic atrophy, which was observed on the FAF image of the right eye (Figure 1B), was indicated by hypo-AF in the macula. The visual field of the right eye was full with a minor deficit in detection of the IVe stimulus (green line, Figure 1C), and an enlarged blind spot scotoma (green with underlay) was detected that reflected the peripapillary atrophy (PPA), visible on the FAF image (Figure 1B). The right (top) eye, though not the left (bottom) eye, showed cystoid macular edema (CME); however, both eyes had an epiretinal membrane (ERM; Figure 1D).

Genetic testing found two potential VUS for patient #1 (*BEST1* c.1003G > C, p.Gly335Arg and *OPA1* c.2797G > T, p.Val933Phe). The *BEST1* missense variant c.1003G > C in exon 9 resulted in change from glycine at position 335 to arginine within the cytoplasmic topological domain (residues 292–585) of the protein. This change was in a highly conserved amino acid (considering 12 species) and was predicted to be pathogenic in 9 out of 10 in silico models. This variant was present in a population study with relatively low frequency (0.0012% in gnomAD). The clinical significance of the identified variant was unknown. Other variants in the same protein domain, p.Thr277Met, p.Asn296Lys, and p.Asp304Asn have been reported in patients affected with best vitelliform macular dystrophy (BVMD) [19]. However, EOG testing resulted in a normal (>2) Arden ratio for the right (2.94) and left (3.04) eyes, suggesting that the heterozygous *BEST1* VUS was not deleterious to RPE function in this patient.

The second VUS identified for patient #1 was in the gene *OPA1* (c.2797G > T; Table 1). The missense substitution occurred at a highly conserved nucleotide (phyloP: 10.00 (−20.0;10.0)) but the amino acid residue was moderately conserved (considering 13 species). There was a small physicochemical difference between valine and phenylalanine (Grantham dist.: 50 (0–215)). These VUS occurred in the protein domains of the Dynamin superfamily and mitochondrial Dynamin-like 120 kDa protein. The VUS affected the 7th amino acid of exon 28. Three algorithms (NNSplice, GeneSplicer, EX-SKIP) observed significant (red) changes for the MUT 3′ splice site of exon 28 as well as an increased likelihood of exon skipping, compared to the WT allele (Appendix A). For these reasons, we hypothesized that the c.2797G > T VUS in *OPA1* would result in altered splicing mechanisms.

To assess whether the *OPA1* VUS results in exon skipping, minigenes containing the WT or the MUT VUS were transiently transfected to COS-7 cells (Figure 1E). RT-PCR analysis for the WT minigene detected the correctly spliced transcript (b) that contained exon 28 (Figure 1F). Splicing of the MUT construct with the c.2797G > T VUS (lanes 2 and 3) resulted in additional bands (a and c) besides the correctly spliced band (b), indicating aberrant splicing (Figure 1F). Sequencing of the band corresponding to the WT product (b) verified that the spliced product included exon 28, whereas the c.2797G > T minigene produced bands at 260 bp (a), 354 bp (b) and ~360 bp (c) (Figure 1F). The MUT minigene products sequenced revealed transcripts with exon skipping (transcript a = 260 bp), inclusion of exon 28 (transcript b = 354 bp), or inclusion of exon 28 with an additional sequence of undetermined origin (Figure 1G), indicating aberrant splicing associated with the *OPA1* VUS (Table 1). If included into the major *OPA1* isoform, the inclusion of extra nucleotides will likely create a frameshift or an unstable transcript predicted to be subject to nonsense-mediated decay (NMD). We concluded that the *OPA1* c.2797G > T variant was likely pathogenic for patient #1.

Patients #2–4 were recently characterized clinically in association with mutations or VUS in the gene *CNGB1* associated with autosomal recessive retinitis pigmentosa (arRP) [20]. SD-OCT imaging was previously presented, which showed that patients #2 and #3 had typical retinal degeneration for patients with RP. Specifically, there was peripheral thinning of the retina with a loss of the inner segment ellipsoid (ISe) band (representing photoreceptors) that were retained in the fovea. Patient #4 displayed only foveal ISe band and CME in the right eye, whereas the left eye SD-OCT image appeared relatively normal, as the degenerative changes were restricted to the inferonasal retina. We also reported the in silico modeling of the VUS [20]. Here we tested three of the VUS (c.2492 + 1G > A, c. 583+2T > C, and c.2305-34G > A; Table 1) that were predicted to affect splicing of the *CNGB1* gene. The *CNGB1* VUS c.2492+1G > A (Figure 2A) was expected (MaxEnt, NNSPLICE, and SSF) to remove (−100%) the donor site for exon 25 (Appendix A). The VUS was known as dbSNP (rs530551814) and had a minor allele frequency (MAF) of 0.000% (1000GENOMES:phase3 frequencies). The consequence of this change was not predictable, but we hypothesized that a skip of exon 25 was prone to be associated with the c.2492+1G > A VUS in the *CNGB1* gene.

The *CNGB1* VUS c.583 + 2T > C substitution was located at the donor site of intron 9. In silico modeling revealed that the VUS abolished the 5′ splice site of exon 9 (SSF, MaxEntScan, NNSPLICE, and GeneSplicer; Appendix A). The VUS was found in dbSNP (rs755036276) and gnomAD (ALL: 0.0012%). The consequence of this change was not predictable, but a skip of exon 9 was likely. We hypothesized that the VUS would result in no splicing (intron inclusion) or if splicing did occur, an alternate donor sight would be used.

The VUS c.2305-34G > A was located 34 bp upstream of exon 24 in the 23rd intron of *CNGB1*. This variant was known as gnomAD (2.1) and dbSNP (rs370223084) with frequencies reported at 0.0097% (ALL) and 0.0001%, respectively. There were several changes predicted by splice algorithms (Appendix A). Notably, the VUS created a new acceptor in intron 23. For these reasons, we hypothesized that the c.2305-34G > A VUS in *CNGB1* would result in intron retention with the 5′ end of exon 23 spliced to the newly created acceptor splice site in the 23rd intron.

To analyze how the *CNGB1* c.2492 + 1G > A or c.583 + 2T > C VUS affected splicing, WT and MUT minigenes, covering exon 25 (Figure 2A) or exons 8–10 (Figure 2B), respectively, and part of the adjacent intron sequences were generated. After transfection into COS-7 cells, the spliced products were analyzed with RT-PCR. The WT *CNGB1* (c.2492 + 1G) minigene resulted in one distinct product of 383 bp (b, Figure 2C; lane 2) and the WT *CNGB1* (c.583 + 2T) minigene produced three bands at 309 (c), 563 (d), and 487 (e) bp (Figure 2C; lane 4). By sequencing, the 383 bp product represented correct splicing of exon 25 (Figure 2D) and the products at 309 bp, 563 bp, and 487 bp corresponded to alternatively spliced transcripts that included exon 9 only (c), exons 9 and 10 (d), or exons 8–10 (e), respectively (Figure 2E). Transfection of the *CNGB1* MUT c.2492 + 1G > A minigene resulted in a single 260 bp (a) product due to exon 25 skipping (Figure 2C). Transfection of the MUT *CNGB1* c.583 + 2T > C minigene resulted in three products: 260 bp (a), 438 bp (f) and 514 bp (g; Figure 2C). The *CNGB1* MUT c.583 + 2T > C minigene products were sequenced, which revealed that the transcripts contained either no *CNGB1* exons (a), only exon 10 (f), or exons 8 and 10 (g), with exon 9 skipped in all observed transcripts (Figure 2E). Thus, we observed exon skipping of the nearest exon (25 or 9) when either the c.2492 + 1G > A or c.583 + 2T > C variants were introduced, respectively (Table 1).

To demonstrate whether the *CNGB1* c.2305-34G > A variant affected splicing, WT and MUT *CNGB1* minigenes were generated (Figure 3A). Transfection of the minigenes resulted in three distinct spliced transcripts of 260 bp (a), 325bp (b) and 412 bp (c) (Figure 3B). Sequencing revealed that the 260 bp product corresponded to skipping of exons 23 and 24 (Figure 3C). The 325bp band represented transcripts containing exon 24, while the 412bp fragment was the result of inclusion of exons 23 and 24 (Figure 3C). The assay was repeated three times using the mCherry-pSPL3 vector and densitometry was performed to quantify the expressed transcripts. The percentage of transcripts that skipped both exons 23 and 24 (transcript a = 260 bp) were found to be higher for the MUT allele (black bar; *p* = 0.007) whereas the products containing either exon 23 only (transcript b = 325 bp) or exons 23 and 24 (transcript c = 412 bp) were lower in the MUT minigene compared with the WT minigene (white bars; *p* = 0.012; *p* = 0.03; Figure 3D, Table 1). These results support a likely pathogenic classification and were responsible for the patients’ arRP.

Patient #5 was diagnosed with Leber congenital amaurosis (LCA). However, no clinical information was available for dissemination. Genetic testing revealed the variants c.929-142G > A and c.930T > A (p.Ser310Arg) in the gene *CLUAP1* (Table 1). The first VUS were located in intron 9 of *CLUAP1* at c.929-142G > A (Figure 4A). These VUS were known to dbSNP (rs1173776498) but were not found in the ClinVar database. The frequency of the VUS in the TOPMED database was 2/125568 (0.0016%). The consequence of this change was not predictable but splicing algorithms (SSF and MaxEntScan) indicated that the c.929-142G > A VUS created a new acceptor splice site (Appendix A). Therefore, we hypothesized that usage of the inappropriate 3′ acceptor splice site would result in the inclusion of part of intron 9 in the expressed transcripts.

The second *CLUAP1* VUS c.930T > A (p.Ser310Arg) variant changed the acceptor site of exon 10. This variant was known as dbSNP (rs1373117341) and has been observed in population databases with a frequency of 0.00040% (gnomAD, 2.1; ALL exomes). The VUS occurred at a weakly conserved nucleotide (phyloP: 1.26 [−20.0;10.0]) and a weakly conserved amino acid (considering 12 species). There was a moderate physicochemical difference between serine and arginine (Grantham dist.: 110 [0–215]). In silico modeling showed a decrease of 25.1% at the acceptor site 2bps upstream including −8.4% (MaxEnt) and −41.7% (NNSPLICE) with scores for the 5′ splice site not recognized (Appendix A).

To determine if the *CLUAP1* c.929-142G > A or c.930T > A VUS affected RNA splicing, WT and MUT minigenes were generated (Figure 4A,B, respectively). The minigenes were transfected into COS-7 cells and the splicing outcomes were analyzed. Both VUS were associated with exon 10 in *CLUAP1*, therefore the WT was the same minigene for each VUS (c.929-142G > A or c.930T > A). The WT *CLUAP1* and the c.929-142G > A minigenes resulted in one distinct product at 368 bp (Figure 4C; lanes 2 and 3, respectively) which corresponded to correct splicing of exon 10 (Figure 4D). Another 442bp fragment was produced by the c.930T > A minigene (Figure 4C; lane 3). Using BLAST and selecting the program to optimize for “somewhat similar sequences”, we determined that the spliced product was the result of using an alternative acceptor site in intron 9 on *CLUAP1*, located 102bp upstream of exon 10 (Figure 4D). The mCherry-pSPL3 vector was used and the transcripts were quantified, which showed a significant increase (>50%) in the WT transcript for MUT c.929-142G > A (*p* = 0.009) and a decrease (>50%) in the WT transcript for MUT c.930T > A (*p* = 0.002) compared to the *CLUAP1* WT minigene (Figure 4E). Thus, both of the VUS in *CLUAP1* identified in a patient with LCA resulted in disruption of normal splicing mechanisms in our minigene assay (Table 1).

## 4. Discussion

One of the current challenges in genetic diagnosis is the verification of VUS. Further measures in genetic diagnosis can be attained when VUS that alter the splicing patterns are evaluated through functional analysis. Splice assays represent a viable choice to assess the effect of VUS on splicing mechanisms in the absence of patient cells, or if the gene of interest is exclusively expressed in the retina.

Pathogenic variations in the *BEST1* gene have been shown to be associated with age-related macular degeneration, BVMD, retinitis pigmentosa, and vitreoretino-choroidopathy [Genetics Home Reference]. Although patient #1 exhibited similar fundus features associated with BVMD, the Arden ratio was within normal limits and the RPE visualized on SD-OCT (Figure 1D) was atypical of bestrophinopathy [21]. A review of the clinical information revealed that the patient had been treated for 20+ years for interstitial cystitis with the drug Pentosan Polysulfate (Elmiron), which is linked to similar maculopathy [22,23] observed for patient #1. Dominant optic atrophy is characterized by central vision loss due to retinal ganglion cell (RGC) degeneration, which typically has an early age of onset that results in severe optic atrophy [24]. However, the disease has incomplete penetrance and variable expression even within families, which ranges from subclinical disease to blindness. Mild or subclinical dominant optic atrophy due to hypomorphic alleles that alter splicing has been associated with *OPA1* (*optic atrophy 1*) [25]. Additionally, an atypical natural history of dominant optic atrophy was recently described in a patient with late onset (62 years) retinal degeneration characterized by acute loss of vision and associated with a dominant mutation in *OPA1* [26]. It has been suggested that missense mutations in *OPA1* cause a more severe phenotype than mutations that alter splicing [25]. The VUS that we evaluated here were missense variants that affected splicing (Figure 1). The VUS segregated with an unaffected daughter, which suggest that the *OPA1* c.2797G > T could be a hypomorphic allele. The daughter (age 35 at genetic testing) may develop optic atrophy in the future. For these reasons, we conclude that the *OPA1* VUS is possibly pathogenic and that the maculopathy found for patient #1 was the result of Elmiron toxicity [22,23].

In this study, we tested two donor site VUS and one acceptor site VUS in the *CNGB1* (*cyclic nucleotide gated channel β 1*) gene. The donor site VUS c.2492 + 1G > A and c.583 + 2T > C were located near the 5′ of exons 25 and 9, respectively, resulting in skipping of these exons in our minigene assay (Figure 2). Exon skipping associated with the c.2492 + 1G > A VUS may result in an in-frame deletion of exon 25, which encodes part of a transmembrane domain of CNGB1, thereby destabilizing formation of a functional CNG channel in the photoreceptor outer segments. Skipping of exon 9 was predicted to result in a frameshift of the coding region, which may introduce a premature stop codon and the resultant amino acid nomenclature may be p.Val179Argfs*82. Since mutations resulting in premature stop codons are known to trigger NMD [27], transcripts in the retina are expected to be affected by NMD in vivo, which may negatively affect channel expression in patient #4 with the c.583 + 2T > C variant in *CNGB1*.

The CLUAP1 (clusterin-associated protein 1) protein plays a central role in photoreceptor ciliogenesis in the vertebrate retina. There are only three mutations listed in ClinVar as pathogenic or likely pathogenic, which include c.817C>T, p.Leu273Phe [28], c.338T > G, p.Met113Arg [29], and c.688C > T, p.Arg230Ter [30]. However, there are 80 VUS that have not been verified for pathogenicity. The two VUS, c.929-142G > A and c.930T > A (p.Ser310Arg), evaluated are located near or in exon 10. Although the c.929-142G > A variant did not result in different splicing, the affected intronic sequence enhancer (ISE) for the protein SRp55 can promote transcription which is what we discovered in our minigene assay. However, we cannot predict whether this occurred in vivo nor deny that transcription efficiency associated with the c.929-142G > A variant was specific to our minigene system. Moreover, due to the limitation of exon trapping experiments, we cannot completely exclude the possibility that in photoreceptors, the VUS may have other effects on splicing. Skipping of exon 10 was found for the c.930T > A variant (Figure 4) which may result in an in-frame loss of 37 residues and an insertion of arginine (p.Ser310_Gly346delinsArg). The consequence of this change on photoreceptors is unknown but it is expected that a significant change in protein folding and tertiary structure to the CLUAP1 protein may occur.

The main strength of the minigene assay is the ability to demonstrate that specific nucleotide changes affect splicing efficiency and mechanisms. However, it is important to note that mutations that affect splicing of an exon in one cell type may not affect splicing in a different cell type because different cell lines can exhibit different alternative splicing patterns due to expression of specific regulatory element proteins needed for the tissue of interest. Additionally, minigene vectors are limited in the size of DNA segments that they can accommodate. The maximum insert size for the pSPL3 vector was reported in a study evaluating an 8 kb insert from the *CNGB3* gene [30]. When analyzing larger segments such as in the *ABCA4* gene, bacterial artificial chromosome clones can accommodate large, multiexon vectors (midigene) to analyze splice variants [31]. The maximum insert size of segments that were cloned into the midigene vector was 11.7 kb. Due to our insert segments being relatively small, we decided to use the minigene assay. Although our minigene assay was appropriate for the analysis of our VUS, a limitation to our study was that we only used one cell line to test our constructs. As more VUS are discovered in our patient population, we will use multiple cell lines, including 661W and ARP19 cells to compare potential splicing defects between cell types. Because most IRD genes are specifically expressed in the retina, a photoreceptor-like cell line may yield a different outcome than what we observed using COS7 cells. Nevertheless, our minigene assay effectively demonstrated that particular nucleotide changes resulted in altered splicing patterns.

In vitro evaluation of six novel VUS in *OPA1*, *CNGB1*, and *CLUAP1* highlights the relevance of pathogenic splicing and increases the genetic diagnostic yield for patients with IRD. Due to emerging gene and cell therapies, the results shown here expand the genotype-phenotype correlations in IRD.

## Figures and Tables

**Figure 1 genes-12-00993-f001:**
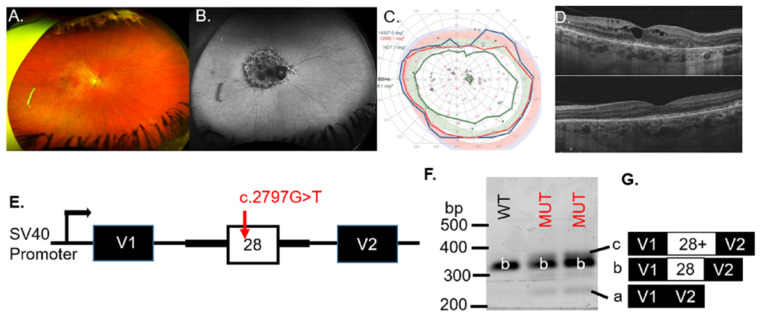
Analysis of *OPA1* VUS c.2797G > T in a patient with (**A**) RPE and degenerative changes, arteriole narrowing, and (**B**) hypo-AF in the macula and around the optic nerve head, illustrated in the right-eye images. (**C**) The visual field of the right eye was full with minor decreased detection of the IVe stimulus (green line) and an enlarged scotoma (green with underlay) at the area corresponding to the peripapillary atrophy (**B**). (**D**) The right (top) eye showed cystoid macular edema on SD-OCT. (**E**) Schematic illustrations of the pSPL3-OPA1 minigene. The OPA1 exon 28, with wild-type (WT) or mutant (MUT) alleles, was cloned between vector exons (V1 and V2). (**F**) Representative gel electrophoresis of RT-PCR products from transfected COS-7 cells. (**G**) WT and MUT transcript content, determined by Sanger sequencing.

**Figure 2 genes-12-00993-f002:**
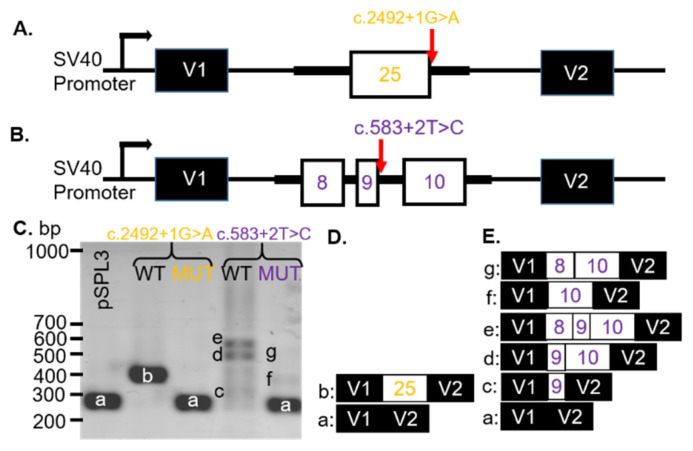
Exon skipping due to *CNGB1* VUS. (**A**) Schematic illustrations of the pSPL3-CNGB1 c.2492 + 1G > A and (**B**) c.583 + 2T > C minigenes. CNGB1 exons (white) and flanking introns (thick black lines) were cloned into the pSPL3 vector (black) with a wildtype (WT) or mutant (MUT) alleles between two pSPL3 exons (V1 and V2). (**C**) Representative gel electrophoresis of RT-PCR products from transfected COS-7 cells. pSPL3 indicates cells that were transfected with a vector containing no gDNA insert. (**D**) WT and MUT transcript content, determined by Sanger sequencing (**E**) a = 260 bp, b = 383 bp, c = 309 bp, d = 563 bp, e = 487 bp, f = 438 bp, g = 514 bp. bp: base pairs.

**Figure 3 genes-12-00993-f003:**
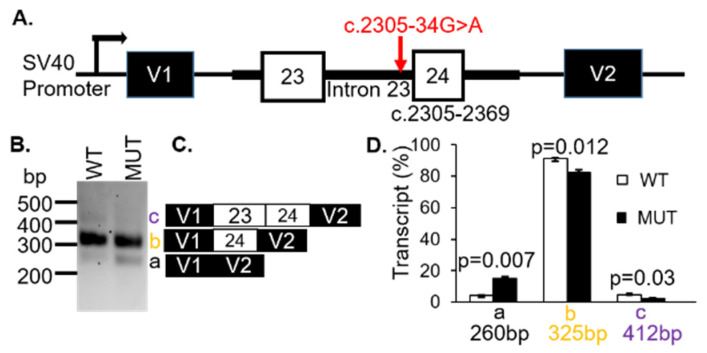
The *CNGB1 c.2305-34G>A* VUS affected splicing efficiency. (**A**) Diagram showing the genomic region amplified from the patient’s genomic DNA, cloned into the pSPL3 vector. (**B**) Representative analysis of mRNA from COS7 cells transfected with wild type (WT) or mutant (MUT) genomic sequences. (**C**) Products identified with Sanger sequencing. (**D**) Transcript (%) relative to total transcripts was significantly different between WT and MUT.

**Figure 4 genes-12-00993-f004:**
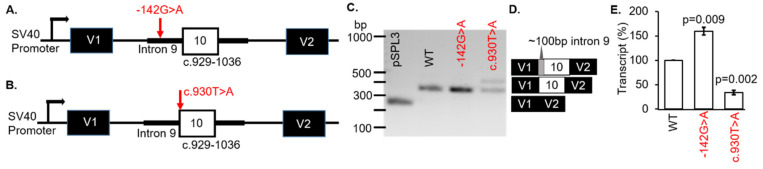
*CLUAP1* VUS disrupted splicing in a minigene splice assay. (**A**,**B**) Diagrams showing the genomic region and location of the variants cloned into the pSPL3 vector. (**C**) Representative RT-PCR analysis from COS-7 cells transi-ently transfected with the single constructs as indicated. (**D**) Sanger sequencing showed vector exons only for the empty vector (V1 + V2), correctly spliced transcripts containing exon 10, and abnormal transcripts that also included ~100 bp of additional sequences. (**E**) The percentage of wild-type (WT) transcript was significantly different for both VUS. V1 and V2, and vector exons 1 and 2, respectively. bp: base pair.

**Table 1 genes-12-00993-t001:** Patient VUS and splice assay outcome.

Patient #	Clinical Diagnosis	Gene	Variant	Outcome
1	Pattern dystrophy	*OPA1*	c.2797G > T (p.Val933Phe)	exon skipping, intron retention, and normal splicing
2	arRP	*CNGB1*	c.2492 + 1G > A (p.?)	exon skipping
3	arRP	*CNGB1*	c.2492 + 1G > A (p.?)	exon skipping
4	arRP	*CNGB1*	c.583 + 2T > C(p.?)/c.2305-34G > A	exon skipping/transcript abundance changed
5	Leber congenital amaurosis	*CLUAP1*	c. c.929-142G > A/c.930T > A (p.Ser310Arg)	transcript abundance changed/intron retention and normal splicing

autosomal recessive retinitis pigmentosa, arRP.

## Data Availability

The data presented in this study are available on request from the corresponding author. The data are not publicly available due to privacy.

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
