# Peer review of "Functional Evaluation of Splicing for Variants of Uncertain Significance in Patients with Inherited Retinal Diseases"

_genes, 2021, doi:10.3390/genes12070993_

Round 1

Reviewer 1 Report

the present paper looking at VUS in selected gene extends the present knowledge on the functional consequences in patients with IRD. The topic is interesting and experiments well designed and performed. A table linking each patient with VUS and functional analysis would be useful.

Author Response

Thank you for your suggestion. We have added a table linking VUS to the patient and outcome. (Table 1)

Reviewer 2 Report

Authors used a minigene assay for functional evaluation of splicing defects in OPA1, CNGB1 and CLUAP1 genes. It’s an interesting work where authors validate mutations using cell lines. I highly recommend accepting this study with subject to minor revisions. Following are my specific comments;

  1. I highly recommend authors to add OCT pictures for all patients in supplementary and also discuss about genotype and phenotype relationship with regard to OCT changes.
  2. I recommend authors to add all results in a tabular form, it would give overall view regarding results.
  3. Authors have used COS-7 cell line for their experiment to understand retinal dystrophy related splicing mutations. I recommend authors to explain why cos-7 cell line has been used for this experiment in last paragraph in introduction.
  4. In discussion, authors should also discuss whether the results obtained from cos7 cell line will vary if arpe19 cell line have been used.
  5. I recommend authors to discuss why functional analysis has been done only for splicing mutations in OPA1, CNGB1 and CLUAP1 genes.
  6. Authors should provide brief information in methods about how the cells were grown and what cell density has been used for transfection.

Author Response

Thank you for your review and recommendations for our manuscript.

  1. SD-OCT was presented previously for patients with CNGB1 mutations. However, we included a statement explain this as well as a description of the findings observed on OCT. We did not have access to SD-OCT imaging for the CLUAP1 patient. (Lines 177-182)
  2. The results were included in a table as suggested. (Table 1)
  3. We added the sentence “By transfecting mammalian cells such as COS7 cells, splicing mechanisms can be evaluated by analyzing the transcripts expressed and processed by the cell’s inherent splicing machinery” at the end of the introduction as suggested (lines 46-48).
  4. We added discussion about different cell lines. Specifically, in the discussion, we added: “The main strength of the minigene assay is the ability to demonstrate that specific nucleotide changes affect splicing efficiency and mechanisms. However, it is important to note that mutations that affect splicing of an exon in one cell type might not affect splicing in a different cell type. This is because of different cell lines can exhibit different alternative splicing patterns due to expression of specific regulatory element proteins needed for the tissue of interest. The limitation to this study is that we only used one cell line to test our minigene constructs. As more VUS are discovered in our patient population, we will use multiple cell lines, including 661W and ARP19 cells to compare potential splicing defects between cell types. Because most IRD genes are specifically expressed in the retina, a photoreceptor-like cell line could yield a different outcome than what we observed using COS7 cells. Nevertheless, our minigene assay was effective to demonstrate that particular nucleotide changes resulted in altered splicing patterns.” (Lines 349-360)
  5. To elaborate on why these VUS were chosen for analysis, we added a sentence to the methods section. It now reads, “The VUS chosen for analysis because they were detected in our patient population and because there was no functional evidence in ClinVar for these VUS.” (Lines 73-74)
  6. As suggested, more information about the transfections was added to the methods section. We added: “Transfections were done using Lipofectamine 3000 transfection reagent (Invitrogen, USA). A 1:2 ratio plasmid DNA:Lipofectamine was used. COS-7 cells were incubated with DNA plasmid-Lipofectamine complex for 6 hours in Opti-MEM® Reduced-Serum Medium (Gibco-ThermoFisher USA). After 6 hours media was changed to DMEM (Dulbecco’s Modified Eagle’s Medium) supplemented with 10% FBS and 1% antibiotic. Cells were collected 48 hours after transfection.” (lines 85-91) and “Transfected cells were collected after 48 hours in regular DMEM supplemented with 10% FBS and 1% antibiotic, and plated at ~ 20 % confluency in DMEM supplemented with 10% FBS and 7 ug/ml blasticidin. COS-7 cells treated with Lipofectamine 3000 but not plasmid were used as a control to follow cell viability with blasticidin.” (100-103)